

# When moult overlaps migration: moult-related changes in plasma biochemistry of migrating common snipe

Patrycja Podlaszczuk[1], Radosław Włodarczyk[1], Tomasz Janiszewski[1], Krzysztof Kaczmarek[2] and Piotr Minias[1]

[1] Department of Biodiversity Studies and Bioeducation, University of Łódź, Łódź, Poland
[2] Department of Electrocardiology, Medical University of Łódź, Łódź, Poland

## ABSTRACT

Moult of feathers entails considerable physiological and energetic costs to an avian organism. Even under favourable feeding conditions, endogenous body stores and energy reserves of moulting birds are usually severely depleted. Thus, most species of birds separate moult from other energy-demanding activities, such as migration or reproduction. Common snipe *Gallinago gallinago* is an exception, as during the first autumn migration many young snipe initiate the post-juvenile moult, which includes replacement of body feathers, lesser and median wing coverts, tertials, and rectrices. Here, we evaluated moult-related changes in blood plasma biochemistry of the common snipe during a period of serious trade-off in energy allocation between moult and migration. For this purpose, concentrations of basic metabolites in plasma were evaluated in more than 500 young snipe migrating through Central Europe. We found significant changes in the plasma concentrations of total protein, triglyceride and glucose over the course of moult, while the concentrations of uric acid and albumin did not change. Total protein concentration increased significantly in the initial stage of moult, probably as a result of increased production of keratin, but it decreased to the pre-moult level at the advanced stage of moult. Plasma triglyceride concentration decreased during the period of tertial and rectrice moult, which reflected depletion of endogenous fat reserves. By contrast, glucose concentration increased steadily during the course of moult, which could be caused by increased catabolism of triglycerides (via gluconeogenesis) or, alternatively, due to increased glucocorticoids as a stress response. Our results suggest that physiological changes associated with moult may be considered important determinants of the low pace of migration typical of the common snipe.

Corresponding author
Piotr Minias,
pminias@biol.uni.lodz.pl,
pminias@op.pl

## INTRODUCTION

Moulting is a process by which the birds maintain feathers in good quality, which improves birds' flight performance and enhances thermoregulation. However, synthesis of feathers is one of the most physiologically costly events in the annual cycle of birds and it requires substantial stores of nutrients in body (*Murphy, 1996*). While the apparent nutrient and energy costs of moult associated with deposition of materials in new feathers may be

relatively mild when compared with the costs of maintenance or reproduction (*Murphy & King, 1992*), the process of moult requires a wide spectrum of metabolic adjustments that are not directly related to plumage synthesis. These additional metabolic processes include recrudescence of the integument, cyclic osteoporosis, and an increased whole-body protein turnover, which may add up to daily energy costs of peak moult exceeding 50% of basal metabolic rate (*Murphy & King, 1992*). In fact, the energy deposited daily as keratins in feathers was estimated to equal only ca. 10% of the energy costs of moult and much higher energy costs were associated with protein metabolism not directly related to keratin synthesis (*Murphy & Taruscio, 1995*).

The biochemical analysis of blood is a technique widely used to indicate avian body condition and to investigate physiological processes during different phases of life. In general, plasma metabolites reflect various aspects of physiological condition and characterize the feeding state of birds. Total protein and triglyceride levels reliably indicate nutrient status of wild and captive birds (*Jenni-Eiermann & Jenni, 1998*; *Jenni-Eiermann, Jenni & Piersma, 2002*; *Albano et al., 2016*), although triglyceride levels may also vary in relation to environmental conditions and stress (*Artacho et al., 2007*; *Ibañez et al., 2015*). Glucose level in plasma decreases during periods of fast and, thus, may serve as an indicator of short-term changes in food intake (*Jenni-Eiermann & Jenni, 1998*; *Totzke et al., 1999*; *Alonso-Alvarez et al., 2002*). Numerous studies indicated that glucose levels are positively correlated with different components of condition or with a broadly-defined individual quality (*Alonso-Alvarez et al., 2002*; *Minias & Kaczmarek, 2013*). High levels of plasma glucose are also associated with increased glucocorticoids as a stress response (*Mondal et al., 2011*), although this relationship may be obscured by the processes of protein catabolism, gluconeogenesis, and insulin regulation (*Remage-Healey & Romero, 2001*; *Cyr et al., 2007*). Plasma concentrations of nitrogenous excretion components, such as uric acid, increases substantially in response to starvation, when tissue proteins are actively mobilized as a source of energy. Plasma concentration of uric acid is a good indicator of condition, especially when individuals have low fat reserves, which rapidly activates protein catabolism during food shortage (*Villegas et al., 2002*). Finally, low albumin concentration may reflect acute diseases and chronic infection or inflammation, which may result from decreased allocation of resources to the immune function (*Hõrak et al., 2002*).

The presented literature shows that changes in blood plasma biochemistry may well serve to evaluate physiological costs of moult. Earlier studies investigated changes in plasma biochemistry during moult in captive birds (*Dolnik & Gavrilov, 1979*; *Murphy & King, 1984*) and other wild-living but flightless birds (*Ghebremeskel et al., 1989*; *Cherel, Charrassin & Challet, 1994*). However, few, if any, papers have examined moult-related changes in plasma biochemistry of wild birds during migration. Most avian species separate moult from other energy-demanding activities, such as migration or reproduction, but several species of birds have been reported to show a moult-migration overlap to a varying degree (*Pérez-Tris et al., 2001*; *Rohwer et al., 2009*), including the common snipe *Gallinago gallinago* (*Podlaszczuk et al., 2016*). Adult common snipe start post-breeding moult at the breeding grounds, as soon as they conclude reproductive activities, and continue moulting during migration. Young common snipe typically begin the partial post-juvenile

moult during their first autumn migration, although probably some individuals can delay moulting until arrival at wintering grounds (*Podlaszczuk et al., 2016*). The post-juvenile moult of the common snipe is more extensive and, thus, more energetically expensive than in other waders, as it includes replacement of body feathers, lesser and median wing coverts, tertials, and rectrices (*Włodarczyk et al., 2008*; *Minias et al., 2010a*). In these respects, the common snipe provides a good opportunity to study moult-related changes in blood plasma biochemistry during a period of serious trade-off in energy allocation between migration and moult. The aim of this study was to determine physiological consequences of moult in migrating common snipe. For this purpose, we measured plasma concentrations of basic metabolites in over half a thousand moulting and non-moulting young common snipe at their final phase of migration through Central Europe.

## METHODS

### Study site and species

Common snipe were captured at the Jeziorsko reservoir (51°40′N, 18°40′E), central Poland, during the autumn migration (04 August–25 September) to the south-west. Jeziorsko reservoir is one of the most important stopover sites for migrating waders and waterfowl in inland Poland, due to the water management policies which ensure considerable seasonal oscillations of water level. In autumn, water level at the reservoir decreases at a constant rate, continuously exposing new areas of mudflats, which provide abundant food resources and attract large flocks of migrating waders. The maximum number of common snipe at the site exceeds a thousand individuals in August (*Janiszewski et al., 1998*).

The common snipe breeds in low Arctic and boreal zones throughout entire Palaearctic, and migrates to the wintering grounds in South-Western Europe (*Cramp & Simmons, 1986*). As indicated by ringing recoveries, common snipe migrating through inland Poland originate mostly from Central Russian populations (Fig. 1; *Minias et al., 2010b*). Although common snipe also breed in Poland and neighbouring Central European countries, there is no evidence that local individuals use Jeziorsko reservoir as a fuelling site prior to autumn migration, as they probably start their migration before the suitable feeding habitats (mudflats) start to appear at the reservoir (usually in early or mid-August). While the common snipe is known to migrate according to the strategy of energy minimization, which is characterized by the low pace of migration and frequent stopovers (*Włodarczyk et al., 2007*), Jeziorsko reservoir is likely to be one of the last staging sites for birds wintering in France and other West-European countries.

### General field procedures

In total, we caught 1007 first-year common snipe during seven migration seasons (2009–2015). Snipe were captured in walk-in traps and mist nets, occasionally with vocal stimulation. All birds were ringed and aged according to plumage (*Kaczmarek et al., 2007*; *Włodarczyk et al., 2008*). The sex of birds was determined either molecularly (in 2009) from blood samples, following protocols developed by *Kahn, John & Quinn (1998)*, or by morphological measurements, using discriminant equations developed for the same migratory population of the common snipe (*Włodarczyk et al., 2011*). For sexing by

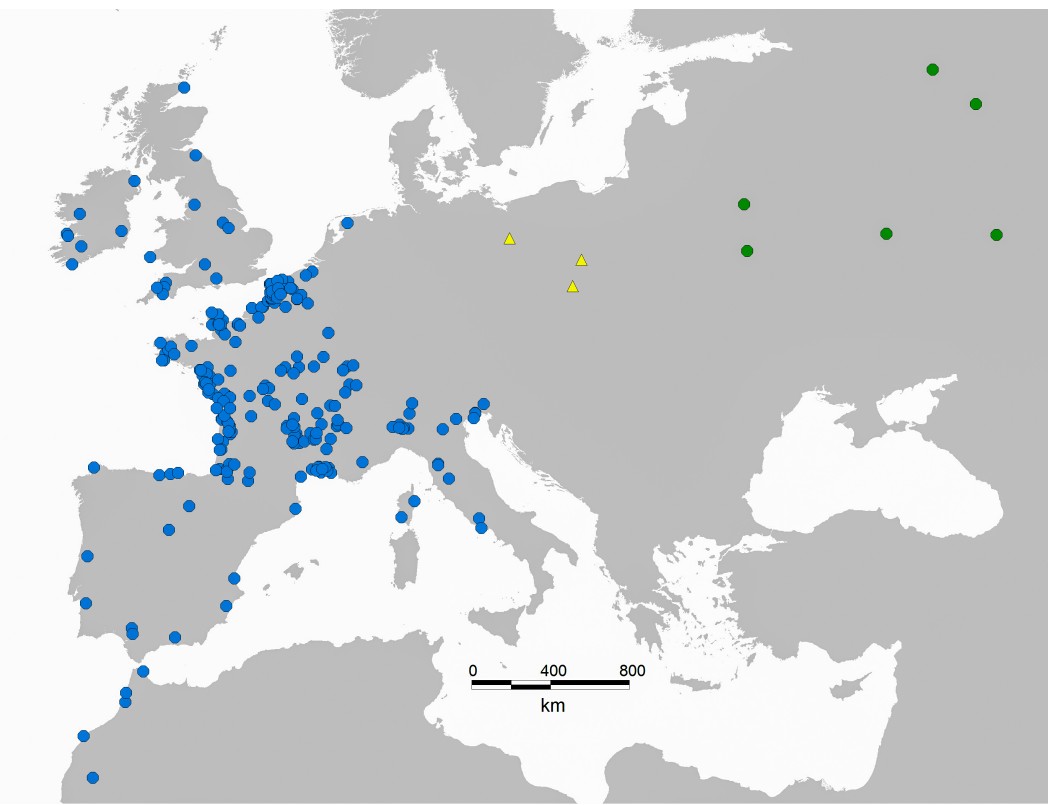

**Figure 1** **Map of ringing recoveries from common snipe migrating through inland Poland.** Ringing sites are marked with yellow triangles, recoveries in spring or summer are marked with green dots (March–September), and recoveries in autumn or winter (September–March) are marked with blues dots. Figure adapted from *Minias et al. (2010b)*.

morphology, bill length and distance between the tips of two outermost rectrices were measured with calipers ($\pm 0.1$ mm) and the vane length of the outermost rectrix was measured with a ruler ($\pm 1$ mm). Fieldwork was performed with permission from the Regional Environmental Protection Directorate in Łódź, Poland. Catching, ringing, and handling birds was performed with permission from the Polish Academy of Sciences, with the approval of the Ministry of Environment in Poland and General Environmental Protection Directorate in Poland.

## Recording moult

In all captured snipe we quantified the stage of post-juvenile moult. During post-juvenile moult snipe change their natal feathers (body feathers, lesser and median wing coverts, tertials, and rectrices) to an adult-type plumage (Fig. 2). Thus, when post-juvenile moult is complete, first-year birds become indistinguishable from adults based on the plumage characteristics. However, few young birds (if any) finish their post-juvenile moult before they reach wintering grounds. Throughout the seven years of study we captured only 43 individuals in fresh (recently moulted) adult-type plumage, most of which were probably adults. All these birds were excluded from analyses. The remaining young birds were classified into one of three moult categories: (1) pre-moult (no feathers moulted); (2)

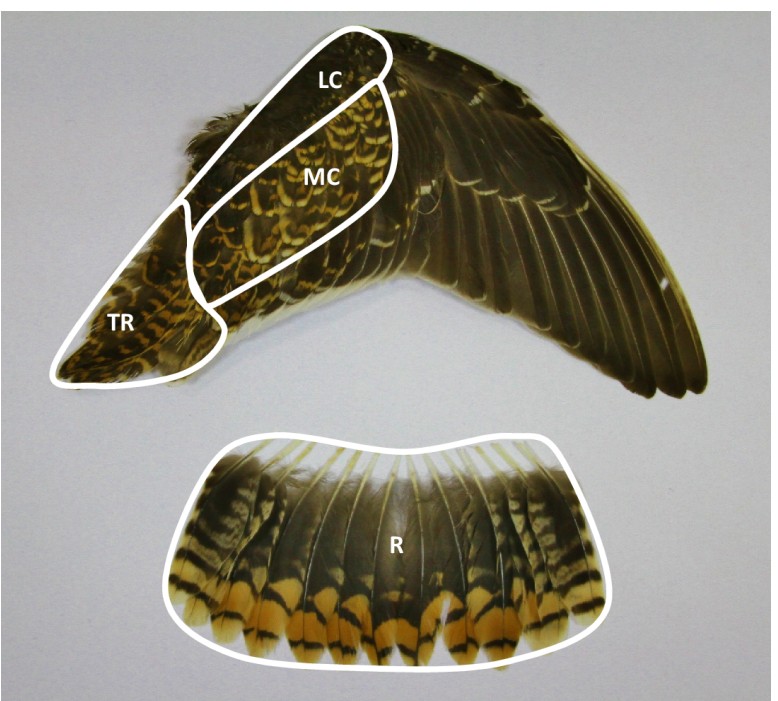

**Figure 2** **The extent of the post-juvenile moult in wing and tail of the common snipe.** Plumage areas marked by white contours are moulted. LC, lesser wing coverts; MC, median wing coverts; TR, tertials; R, rectrices.

initial stage of moult (only body feathers and wing coverts in active moult); (3) advanced stage of moult (tertials or rectrices in active moult). Moult progress was also quantified in more detail for birds that moulted tertials or rectrices. For this purpose, each tertial ($n = 8$) and rectrix ($n = 14$) was given a moult score according to the feather scoring system developed by *Ashmole (1962)*, where: 0—old feather, 1—old feather missing or a new feather in a pin, 2—new feather up to one third grown, 3—new feather between one and two thirds grown, 4—new feather more than two thirds grown, 5—new feather fully developed. A sum of all moult scores for individual feathers was used as a general moult score (max. 110, when all tertials and rectrices were renewed).

## Plasma biochemistry

About 50% of captured young snipe ($n = 538$ individuals) were selected for plasma biochemistry measurements. Between 20 and 40 µl of blood was collected from the ulnar vein of each bird into heparinized capillary tubes. Blood sampling was performed with permission from the Local Bioethical Commission in Łódź, Poland. Samples were centrifuged at 6,000 rpm for 5 min within an hour of collection to separate plasma from blood cells, and kept at −20 °C until analysis. Plasma metabolite concentrations (total protein, albumin, triglycerides, glucose, and uric acid) were analysed with a spectrophotometer (BTS-330; BioSystems Reagents & Instruments, Barcelona, Spain) using commercial kits of the same manufacturer (BioSystems Reagents & Instruments, Barcelona, Spain). All analyses were conducted according to the manufacturer protocols

**Table 1  Numbers of young common snipe in which different plasma parameters were analysed at three stages of the post-juvenile moult.**

| Plasma parameter | Moult stage | | |
|---|---|---|---|
| | **Before** | **Initial** | **Advanced** |
| Total protein | 299 | 171 | 58 |
| Triglycerides | 267 | 146 | 49 |
| Glucose | 213 | 103 | 37 |
| Albumin | 191 | 96 | 35 |
| Uric acid | 75 | 37 | 21 |

using the following methods: total protein (biuret reaction), albumin (bromocresol green), triglycerides (glycerol phosphate oxidase/peroxidase), glucose (glucose oxidase/peroxidase), and uric acid (uricase/peroxidase). Absorbance of each sample was measured in a flow cuvette against a blank reagent at the following wave lengths: 500 nm (glucose, triglycerids), 520 nm (uric acid), 545 nm (total protein), and 630 nm (albumin). Run-to-run repeatability (R) and linearity limits (LL) were specified as follows: total protein (R: 1.85%; LL: 150 g/L), albumin (R: 1.90%; LL: 70 g/L), triglycerides (R: 2.15%; LL: 600 mg/dL), glucose (R: 2.3%; LL: 500 mg/dL), and uric acid (R: 2.00 %, LL: 25 mg/dL). The applied biochemical methods followed the standard methodology used in avian studies (e.g., *Artacho et al., 2007*). Since the amount of plasma collected from each birds was often not sufficient to measure all five plasma biochemistry parameters, sample sizes for each parameter are different (Table 1). Distributions of all plasma metabolite concentrations were reasonably close to normal (skewness: 0.08–0.69) and thus were not transformed.

## Statistical analyses

Differences in plasma biochemistry parameters between consecutive stages of post-juvenile moult were analysed with general linear models (GLMs), separately for each parameter. In each model, we controlled for the effects of sex, year, date of capture (Julian day), and hour of capture. Date was standardized to equal unit variances within each season ($z = \frac{x-\mu}{\sigma}$, where $\mu$ is the mean date of capture in a given season and $\sigma$ is the standard deviation of capture date in a given season) to account for annual variation in the timing of migration. For birds at the advanced stage of moult, we also used GLMs to investigate the effect of moult score on plasma metabolite concentrations. In these models, the general moult score calculated for tertials and rectrices was entered as a covariate. To obtain more parsimonious reduced models, we removed non-significant ($p > 0.15$) predictors from initial full models. All statistical analyses were performed with Statistica 10.0 (StatSoft, Tulsa, OK, USA). All values are presented as means ± SE.

## RESULTS

43.7 % of young common snipe showed signs of post-juvenile moult ($n = 538$). Most moulting snipe (74.9%, $n = 235$) were at the initial stage of moult, while the remaining 25.1% were at the advanced stage of moult.

**Table 2  Total plasma protein concentration in relation to the stages of post-juvenile moult and confounding variables in young common snipe migrating through central Poland.** Reduced model $R^2 = 0.41$ ( $F_{10,517} = 35.85$, $p < 0.001$). Significant predictors are marked in bold.

| Factor | F | p |
|---|---|---|
| Full model | | |
|     Moult stage | **3.46** | **0.032** |
|     Sex | 2.30 | 0.13 |
|     Year | **7.47** | **<0.001** |
|     Date | 1.51 | 0.22 |
|     Hour | **9.68** | **0.002** |
| Reduced model | | |
|     Moult stage | **3.13** | **0.045** |
|     Sex | 2.12 | 0.15 |
|     Year | **7.54** | **<0.001** |
|     Hour | **9.14** | **0.003** |

Plasma concentrations of total protein and glucose differed significantly between the consecutive stages of post-juvenile moult (Tables 2 and 3). Total protein concentration was significantly higher at the initial stage of moult ($35.57 \pm 0.52$ g/l) when compared to the pre-moult stage ($33.33 \pm 0.42$ g/l; Tukey test: $p < 0.001$; Fig. 3A) and to the advanced stage of moult ($32.62 \pm 0.90$ g/l; Tukey test: $p = 0.011$). There was no significant difference in the total protein concentration between the pre-moult and advanced-moult stages (Tukey test: $p = 0.67$; Fig. 3A). By contrast, glucose concentration was higher at the advanced stage of moult than during the pre-moult stage ($511.6 \pm 20.6$ mg/dl vs. $454.8 \pm 7.1$ mg/dl.; Tukey test: $p = 0.039$; Fig. 3B). Snipe at the initial stage of moult had an intermediate concentration of glucose (Fig. 3B). Other plasma parameters showed no variation with the moult stage (Table 4). Only triglyceride concentration in plasma changed with the moult score of snipe that moulted tertials or rectrices ($F_{1,61} = 4.10$, $p = 0.047$), and it significantly decreased during moult of tertials and rectrices ($\beta = -0.29 \pm 0.14$; Fig. 4). The other plasma parameters (total protein, albumin, glucose, and uric acid concentrations) showed no variation related to the moult score of tertials and rectrices (all $p > 0.05$).

## DISCUSSION

Concentrations of total protein, triglycerides and glucose in plasma changed significantly during the post-juvenile moult of the common snipe. At least some of these changes in blood plasma biochemistry are likely associated with the use of energy and nutrients during plumage synthesis or during other moult-related metabolic processes which greatly contribute to the overall costs of moult (e.g., vascularization of integument or alterations to bone metabolism; *Murphy & King, 1992*).

Total protein plasma concentration increased significantly in the initial stage of moulting but fell later during the advanced stage of feather replacement, returning to the low pre-moult level. Snipe have probably the highest protein demand at the beginning of moult, due to the rapid acceleration of keratin synthesis for feather production and other metabolic

**Table 3** **Plasma glucose concentration in relation to the stages of post-juvenile moult and confounding variables in young common snipe migrating through central Poland.** Reduced model $R^2 = 0.16$ ($F_{8,344} = 7.96$, $p < 0.001$). Significant predictors are marked in bold.

| Factor | F | p |
|---|---|---|
| Full model | | |
| Moult stage | **3.60** | **0.028** |
| Sex | 0.41 | 0.52 |
| Year | **14.21** | **<0.001** |
| Date | **4.59** | **0.033** |
| Hour | 3.23 | 0.07 |
| Reduced model | | |
| Moult stage | **3.74** | **0.025** |
| Year | **14.35** | **<0.001** |
| Date | **4.82** | **0.029** |
| Hour | 3.38 | 0.07 |

**Table 4** **Plasma concentrations of albumin, triglycerides, and uric acid in relation to the stages of post-juvenile moult and confounding variables in young common snipe migrating through central Poland.** Significant predictors are marked in bold.

| Factor | Albumin | | Triglycerides | | Uric acid | |
|---|---|---|---|---|---|---|
| | F | p | F | p | F | p |
| Moult stage | 1.42 | 0.24 | 0.12 | 0.89 | 0.44 | 0.65 |
| Sex | 1.58 | 0.21 | 0.09 | 0.77 | 0.01 | 0.92 |
| Year | **9.01** | **<0.001** | **10.23** | **<0.001** | 0.30 | 0.58 |
| Date | 0.02 | 0.88 | 0.27 | 0.60 | **22.03** | **<0.001** |
| Hour | **15.77** | **<0.001** | **3.87** | **0.049** | 2.50 | 0.12 |

processes associated with early phases of moult, such as vascularization of the active feather follicles, pulp formation, and an increase of erythrocytes (*DeGraw & Kern, 1985*; *Murphy & King, 1992*). It has been shown that deposition of protein as keratins of feathers may equal a quarter or more of the total protein mass of the bird (*Newton, 1968*; *Murphy & Taruscio, 1995*; *Roman et al., 2009*). Production of keratin depends largely upon sulphur containing amino acids (cysteine and cystine), which, thus, may be critical for plumage synthesis. For example, *Murphy & King (1984)* showed that moulting white-crowned sparrows *Zonotrichia leucophrys gambelii* require large amounts of glutathione, which primarily consists of sulphur containing amino acids. However, besides playing a role in feather synthesis, plasma proteins have a variety of immunological and transport functions and are important indicators of nutritional state and health of a bird (*Jenni-Eiermann & Jenni, 1996*). Plasma proteins also carry a range of metabolites (*Jenni-Eiermann & Jenni, 1996*). Reduction of total protein content is an indicator of many pathological changes (malnutrition), as proteins contribute to a pool of amino-acids for protein synthesis and can act as a source of energy (*Jenni-Eiermann & Jenni, 1996*).

Our findings are similar to those of *Dolnik & Gavrilov (1979)* who found that total protein level increased at the initial stage of moulting in the chaffinch *Fringilla coelebs*,

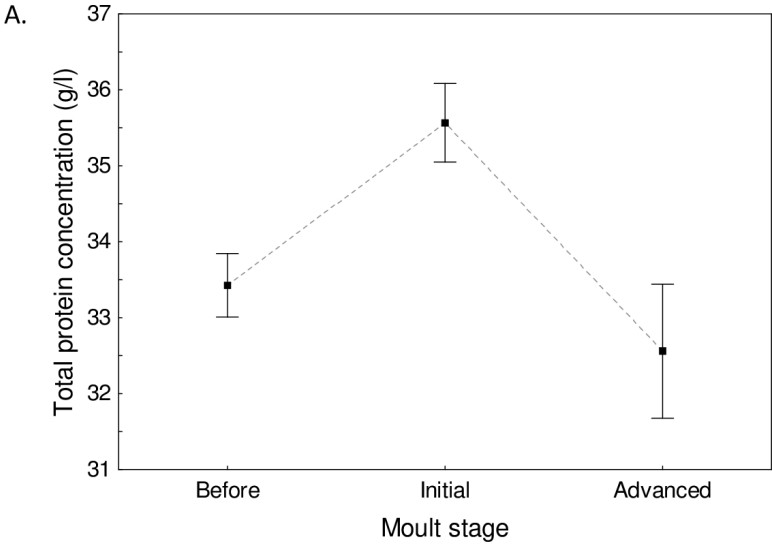

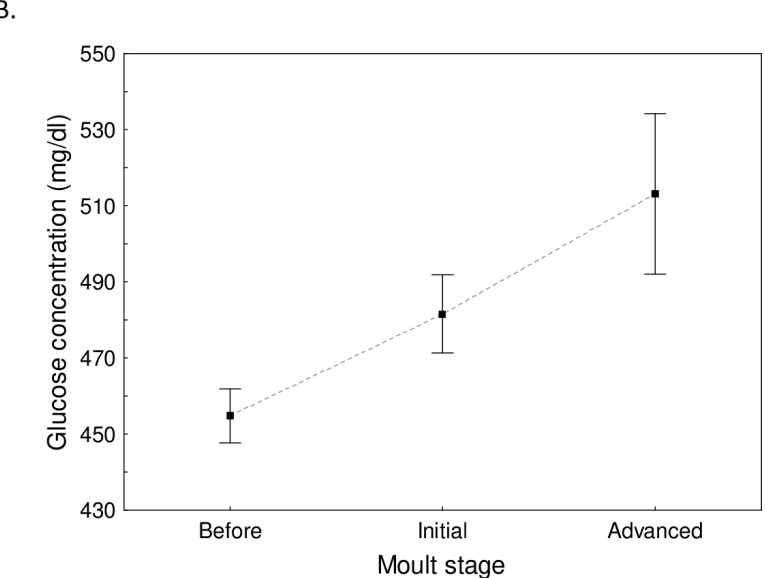

**Figure 3** Changes in plasma concentrations of total protein (A) and glucose (B) between the consecutive stages of post-juvenile moult in young common snipe migrating through central Poland. Means ± SE are presented.

which was due to intensive synthesis of protein as material for new feather production. This initial rise was followed by a decrease over the next stages of moult, similarly as in our study. A decrease in total protein concentration during moult was also recorded in seabirds (*Work, 1996*), passerines (*Newton, 1968*; *DeGraw & Kern, 1985*), ducks and geese (*Driver, 1981*; *Roman et al., 2009*). Other studies showed that the level of total protein was significantly higher after moult than during feather replacement (*Thompson & Drobney, 1996*). Nevertheless, *Ghebremeskel et al. (1989)* found total plasma protein to be significantly lower in the post-moult than the pre-moult stage in rockhopper *Eudyptes crestatus* and Magellanic penguins *Spheniscus magellanicus*. Species vary in their baseline protein level

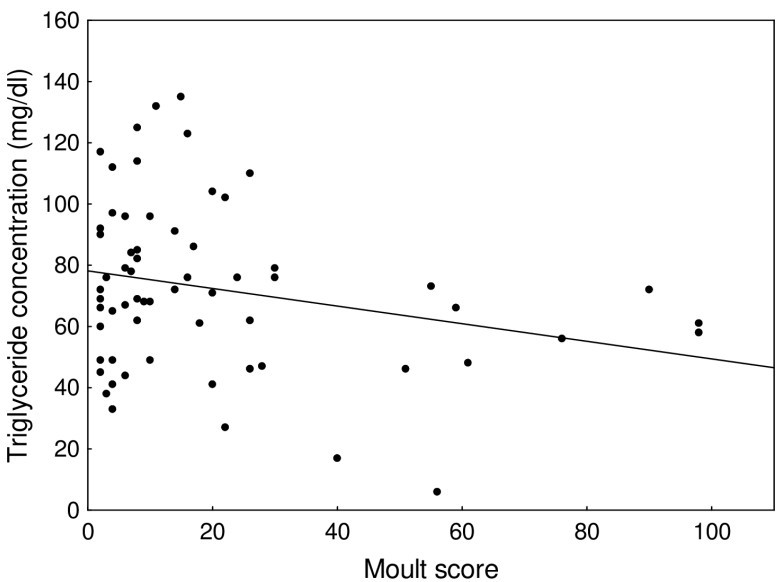

**Figure 4 Changes in plasma triglyceride concentration with moult score of young common snipe in the advanced stage of post-juvenile moult.** The line indicates a fitted regression ($y = -0.29^*x + 78.13$; $R^2 = 0.063$).

and this may result from variations in the supply of amino acids and energy. Most species rely mostly on their diet to meet the growing demand for protein during moulting, but some birds, such as penguins, which do not feed during moult, use endogenous nutrients to synthesize feathers (*Cherel, Charrassin & Challet, 1994*). While it remains unknown whether the common snipe primarily use endogenous or exogenous nutrients for feather synthesis, it was found that snipe depend on endogenous energy from adipocyte cells during moult period (*Minias et al., 2010a*). The decreased levels of plasma total protein observed during the final stages of moult result from an ongoing protein accumulation in feathers or muscles, as well as from less intensive synthesis in the liver (*Roman et al., 2009*). At the advanced stage of moult, some proteins obtained with food could be also catabolized into amino acids and keto acids, and then used primarily as energy or for synthesis of fatty acids (*Artacho et al., 2007*).

Plasma triglycerides are a well-known indicator of malnutrition or fasting, and their concentration decreases rapidly even during overnight fasting (e.g., *Jenni-Eiermann & Jenni, 1996*; *Jenni & Schwilch, 2001*; *Jenni-Eiermann, Jenni & Piersma, 2002*). We found that plasma triglyceride levels decreased in moulting common snipe, which is consistent with previous findings that fat reserves of snipe decreased by ca. 50% between the initial and final stages of the post-juvenile moult (*Minias et al., 2010a*). The decreasing plasma triglyceride level observed during moult is probably an indicator of increasing problems with food supply. To satisfy high energy demand, snipe rely on their fat reserves (*Minias et al., 2010a*) and probably on catabolised protein obtained from dietary sources. Birds catabolise fat reserves to compensate for energy deficiencies in food intake, which is especially likely during such energy-demanding processes as moult (*Jenni-Eiermann & Jenni, 1996*; *Klasing, 1998*; *Jenni & Schwilch, 2001*; *Jenni-Eiermann, Jenni & Piersma, 2002*; *Artacho et al., 2007*).

A number of studies showed that the level of metabolized energy increases during the initial stages of moult, but decreases in the next phases of moult and finally settle at a level below initial values upon moult completion (*Newton, 1968*; *Myrcha & Pinowski, 1970*; *Dolnik & Gavrilov, 1979*; *Jenni-Eiermann & Jenni, 1996*; *Artacho et al., 2007*).

In contrast to triglycerides, plasma glucose concentration in the common snipe steadily increased from the start of the moult until its advanced stage. Glucose is the main product of the carbohydrate metabolism and it is obtained from the diet. Some studies indicate that good body condition is associated with increased glucose level (*Minias & Kaczmarek, 2013*). A decrease in glucose level in birds could be an indicator of short fasting periods (*Jenni-Eiermann & Jenni, 1994*; *Jenni-Eiermann & Jenni, 1997*); however, in some species plasma glucose concentration negatively correlated with body mass (*Kaliński et al., 2014*). During starvation, glucose is produced from stored glycerol and amino acids or by gluconeogenesis (*Herzberg et al., 1988*) and may also occur as a stress-induced hyperglycaemia with increased glucocorticoids (*Remage-Healey & Romero, 2001*).

There are two likely explanations for the increasing levels of plasma glucose during moult in the common snipe. First, snipe use their fat reserves during moult (*Minias et al., 2010a*), which is supported by decreasing plasma triglyceride concentrations and, thus, the increasing glucose level may be an effect of the catabolism of triglycerides, stored in adipocyte cells. During lipolysis, the triglycerides are split into monoacylglycerol units which are converted to free fatty acids and glycerol. Glycerol can be then metabolised into glucose by conversion into dihydroxyacetone phosphate and then into glyceraldehyde 3-phosphate in the process of gluconeogenesis (*Herzberg et al., 1988*). Consequently, we cannot exclude that increasing catabolism of fat may simultaneously elevate plasma glucose levels during moult.

The second reason for increasing plasma glucose concentration may be associated with elevated levels of corticosteroids. Glucocorticoids increase glucose level by working as an insulin antagonist and stimulating lipolysis in adipose tissue, which results in an increase in plasma free fatty acids and glycerol levels (*Remage-Healey & Romero, 2001*; *Ramenofsky, 2011*). Several studies have shown that glucocorticoid activity is associated with migration (*Landys, Ramenofsky & Wingfield, 2006*; *Ramenofsky, 2011*) and high levels of plasma corticosterone have been well documented in many long-distance migrants (*Falsone, Jenni-Eiermann & Jenni, 2009*; *Landys-Ciannelli et al., 2002*; *Reneerkens et al., 2002*). Thus, it seems likely that migrating young common snipe may show higher levels of corticosterone required for the maintenance of migratory condition (*Ramenofsky, Piersma & Jukema, 1995*; *Holberton, 1999*; *Landys-Ciannelli et al., 2002*; *Reneerkens et al., 2002*; *Falsone, Jenni-Eiermann & Jenni, 2009*). On the other hand, there is no agreement on how corticosterone level is affected by moult. While baseline and stress-induced levels of corticosterone were lower during moult in the common starlings *Sturnus vulgaris* (*Romero & Remage-Healey, 2000*), some other studies suggested that corticosterone suppression is not a prerequisite for synthesis of high-quality feathers (*Buttemer, Addison & Astheimer, 2015*). Regardless of the mechanism responsible for plasma glucose regulation in moulting common snipe, both pre-moult and moult levels of plasma glucose in snipe were very high when compared to glycemic levels in other bird species (*Prinzinger & Misovic, 1994*; *Beuchat & Chong,*

*1998*). This suggests that plasma glucose concentration in moulting snipe was above the threshold of glycemic requirement and may not be indicative of catabolic compromise.

In conclusion, our study indicates significant changes in blood plasma biochemistry during the post-juvenile moult in the common snipe. These changes, which indicate high nutritional and physiological costs of moult, might be among the primary determinants for the low pace of migration in this species. The common snipe minimizes energy expenditure during autumn migration, a strategy characterized by low refuelling rates, accumulation of small fat reserves, and migrating by short migratory "hops" between a large number of stopover sites (*Włodarczyk et al., 2007*). Our results suggest that physiological changes associated with moult and a trade-off in energy allocation between moult and migration may prevent the common snipe from adopting migration strategy of energetically-expensive long-distance migratory flights.

## ACKNOWLEDGEMENTS

We would like to thank all participants of fieldwork at Jeziorsko reservoir, especially Tomasz Iciek, Anna Piasecka, and Przemysław Wylegała. We also thank Magdalena Remisiewicz and an anonymous reviewer for constructive comments on the earlier drafts of the manuscript.

### Funding

The authors received no funding for this work.

### Competing Interests

The authors declare there are no competing interests.

### Author Contributions

- Patrycja Podlaszczuk performed the experiments, contributed reagents/materials/analysis tools, wrote the paper, prepared figures and/or tables, reviewed drafts of the paper.
- Radosław Włodarczyk performed the experiments, reviewed drafts of the paper.
- Tomasz Janiszewski and Krzysztof Kaczmarek conceived and designed the experiments, performed the experiments, reviewed drafts of the paper.
- Piotr Minias conceived and designed the experiments, performed the experiments, analyzed the data, contributed reagents/materials/analysis tools, reviewed drafts of the paper.

### Animal Ethics

The following information was supplied relating to ethical approvals (i.e., approving body and any reference numbers):

Local Bioethical Commission in Łódź, Poland (no. Ł/BD/278)

General Environmental Protection Directorate in Poland and Ministry of Environment in Poland (nos. DOPozgiz-4200/III-173/622/09/ls—DZP-WG.6401.03.36.2015.km).

## Field Study Permissions

The following information was supplied relating to field study approvals (i.e., approving body and any reference numbers):

Regional Environmental Protection Directorate in Łódź, Poland (nos. RDOS-10-WPN.I-6630-12/09/db; RDOS-10-WPN.I-6630-23-10/kb; WPN-I.6205.4.2011HG; WPN.6205.13.2012.DB.4; WST-SI.6205.6.2013.MJ; WST-SI.6205.6.2014.MJ; WPN.6205.63.2015.HG).

## Data Availability

The raw data has been supplied as a Supplementary File.

## Supplemental Information

Supplemental information for this article can be found online at http://dx.doi.org/10.7717/peerj.3057#supplemental-information.

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
