# Peer review of "When moult overlaps migration: moult-related changes in plasma biochemistry of migrating common snipe"

_PeerJ, doi:10.7717/peerj.3057_

## Round 0.1 · original submission · Major Revisions

Two referees have provided comments which will help the revision of your manuscript. Referee 1 particularly wants more background information or better explanations about the system from previous work.

Reviewer 1 ·

Basic reporting

Concern over "self-contained"
In a previous study, the authors showed that Common Snipes represent the energy minimizer strategy that birds initiate moult on the breeding grounds that can be monitored at a major stopover site – the Jeziorsko Reservoir. They use this information to guide the current study of juvenile common snipe however, a more thorough description of the migratory strategy in the current manuscript would be beneficial as well as a discussion of whether this is typical of adults as well. Also do birds complete moult before departing or are they progressing on with autumn migration and moult simultaneously? Another question that was raised if they initiate moult on the breeding grounds why not complete it there? Can this initial phase of the autumn migration be considered a molt migration? Are there sufficient resources present at the reservoir to support a molt or again is this just on of many stops the birds make?

Experimental design

No Comment

Validity of the findings

For many birds, postnuptial includes not only feather replacement but also components of blood, bone, integument, organs as well etc. Although this may not be the complete case for a post juvenile molt consideration of the fuller spectrum of requirements for molt should be considered. Are such replacements part of the Snipe post juvenile moult?
Line 237, caution is recommended over the authors’ speculation that elevated levels of glucose may be a result of elevated glucocorticoid - corticosterone. However, for most migrants during moult, corticosterone levels are generally lower than at other stages of the annual cycle given the complications of elevated corticosterone and feather production. Also references (Holberton, 1999 and Ramenofsky et al 1995) were taken from birds either prior to or during migration but not from moulting birds, so provide minimal support for the author’s assertions. Figure 2 lower panel does show an elevation of glucose over the period of moult but these levels are very high as is typical of glycemic levels of birds. So one could speculate that these are all above a threshold of glycemic requirement and may not be indicative of catabolic compromise.

Additional comments

It is commonly accepted that the cost of moult is high and so occurs at times that do no interfere with other energetically demanding stages such as breeding and migration. That being the case birds can either moult at times outside the breeding/migration windows or interrupt a stage to engage in a molt or slow moult down to such a point that the costs of replacements are reduced thus not intruding upon current energetic output of a stage such as migration. In this study, the authors report on the post-juvenile moult in common snipe recorded during autumn migration at a stopover site. To investigate the investiture of energy and fuels, authors present their findings of blood biochemistry (plasma metabolites) during autumn migration. The authors propose to determine the physiological costs in terms of blood chemistry of initiating post-juvenile moult during autumn migration to determine trade off of energy allocation during migration.

Specific points:
Methods, Map of breeding, moulting and wintering areas would improve the understanding of the project. As described in a previous work, snipes are energy minimizers so take numerous stopovers en route. Is the reservoir one of many stops and do the snipes continue to moult as they move toward the wintering grounds?

Statistics, were variables tested for normality if so and if some were nonnormal how were they treated?

Line 39, common referring to “increase tissue content” is unclear what is meant by this?
Line 40, lot not lots
Line 54, Note that plasma glucose is not always increase with elevated corticosterone associated with chronic stress as a results of protein catabolism and gluconeogenesis and relationship with insulin but see (Cyr et al. 2007, General and Comparative Endocrinology 154:59-66 and Remage-Healey and Romero, Am. J. Reg. Int. Comp. Physiol. (2001) 281:994).
Line 120 – no mention of time requirements for collection of blood samples for biochemistry for blood metabolites. Some have noted that metabolites increase in relation to stress of capture and handling restraint, were time series conducted to detect changes in blood metabolite levels?
Line 130, descriptions of the biochemical assays too brief more info required for assessment, information about the assays, name of the kits and dynamics of each assay, curves, sensitities, ranges of values etc. Without this information it is difficult to judge the quality of the assays.

Line 138, How was date standardized with each season?

Line 170 -, Results show a peak of total protein at the initiation stages of moult. However feathers production requires production/deposition of keratin that depend largely upon sulfur containing amino acids (cysteine and cistine )are critical (Murphy and King 1984, Condor 86(3):324 showed that molting WCS require glutathione consisting largely of sulfur containing amino acids. Although it would be interesting to know these specific measurements particularly during the initial stages of moult, discussion of these well known specfics would bolster the information base of the manuscript.

Line 209, Authors state that fat catabolism compensates for energy deficiencies but these birds are migrating which relies on lipid and protein fat.

·

Basic reporting

The way of reporting the findings in English, and the clarity of the presentation can be improved (see my numerous comments in the attached msc). The first paragraph of Discussion is repetitive with the Results. But the Discussion and other part in of the text can be improved and edited quite easily, for better clarity. Table captions can also be made more clear, and self-explanatory, without need to look for explanation in the text. Otherwise the structure of the paper is standard and clear, figures and table are relevant and clearly presented.

Experimental design

The experimetal design, both in the field and in laboratory, is appropriate and well suited to the presented results.

Validity of the findings

The results are strong, laboratory methods are relevant, and the statistical analyses are sound. The source data are provided in Supplementary materials and clearly presented. The findings repeat some methods and findings from other species, especially passerines. But the 'replication' of checking the levels of metabolites in birds blood is well justified in this paper by applying the methods to a migrant wader, a group that has not been well studied in this respect, and to an exceptional case of a migrant wader that overlaps moult with migration. The authors then used a unique opportunity they had to apply methods and approach found in other studies, to study and discuss the trade-off in energy allocation between moult and migration.

Additional comments

Edit the text carefully, hope that my comments will be useful.

---

## Round 0.2 · Minor Revisions

The referee has added some minor isses and a comment about an unresolved issue. I have added some more minor typos etc onto the PDf and uploaded again (so you will get two versions, one with referee comments and another with referee and my comments together). Looking forward to receiving revision.

·

Basic reporting

The reporting is clear and professional, with suitable literature references.The clarity of reporting has improved in the resubmitted version. The results meet hypotheses well.

Experimental design

Good

Validity of the findings

Sound and well presented results, well discussed. The findings are interesting and worth publishing.

Additional comments

I added a few minor editing comments, which may help to improve the final manuscript.

---

## Round 0.3 · accepted · Accept

Thank you for your corrections.